# Biomaterials Based on Carbon Nanotube Nanocomposites of Poly(styrene-*b*-isobutylene-*b*-styrene): The Effect of Nanotube Content on the Mechanical Properties, Biocompatibility and Hemocompatibility

**DOI:** 10.3390/nano12050733

**Published:** 2022-02-22

**Authors:** Maria A. Rezvova, Pavel A. Nikishau, Miraslau I. Makarevich, Tatiana V. Glushkova, Kirill Yu. Klyshnikov, Tatiana N. Akentieva, Olga S. Efimova, Andrey P. Nikitin, Valentina Yu. Malysheva, Vera G. Matveeva, Evgeniia A. Senokosova, Mariam Yu. Khanova, Viacheslav V. Danilov, Dmitry M. Russakov, Zinfer R. Ismagilov, Sergei V. Kostjuk, Evgeny A. Ovcharenko

**Affiliations:** 1Department of Experimental Medicine, Research Institute for Complex Issues of Cardiovascular Diseases, 650002 Kemerovo, Russia; bio.tvg@mail.ru (T.V.G.); klyshnikovk@gmail.com (K.Y.K.); t.akentyeva@mail.ru (T.N.A.); matveeva_vg@mail.ru (V.G.M.); sergeewa.ew@yandex.ru (E.A.S.); khanovam@gmail.com (M.Y.K.); ov.eugene@gmail.com (E.A.O.); 2Research Institute for Physical Chemical Problems, Belarusian State University, 220006 Minsk, Belarus; nikishau@bsu.by (P.A.N.); miraslau.makarevich@gmail.com (M.I.M.); kostjuks@bsu.by (S.V.K.); 3Department of Chemistry, Belarusian State University, 220006 Minsk, Belarus; 4Institute of Coal Chemistry and Material Science, Federal Research Center of Coal and Coal Chemistry SB RAS, 650000 Kemerovo, Russia; efimovaos@mail.ru (O.S.E.); nikitinandreyp@yandex.ru (A.P.N.); v23091@yandex.ru (V.Y.M.); zinfer1@mail.ru (Z.R.I.); 5Research Laboratory for Processing and Analysis of Big Data, Tomsk Polytechnic University, 634050 Tomsk, Russia; viacheslav.v.danilov@gmail.com; 6Institute of Fundamental Sciences, Kemerovo State University, 650000 Kemerovo, Russia; dm.russakov@gmail.com; 7Institute for Regenerative Medicine, Sechenov First Moscow State Medical University, 119991 Moscow, Russia

**Keywords:** single-walled carbon nanotubes, polymer nanocomposites, biocompatible polymers, interphase interaction, SIBS, cardiovascular medical devices

## Abstract

Nanocomposites based on poly(styrene-*block*-isobutylene-*block*-styrene) (SIBS) and single-walled carbon nanotubes (CNTs) were prepared and characterized in terms of tensile strength as well as bio- and hemocompatibility. It was shown that modification of CNTs using dodecylamine (DDA), featured by a long non-polar alkane chain, provided much better dispersion of nanotubes in SIBS as compared to unmodified CNTs. As a result of such modification, the tensile strength of the nanocomposite based on SIBS with low molecular weight (M_n_ = 40,000 g mol^–1^) containing 4% of functionalized CNTs was increased up to 5.51 ± 0.50 MPa in comparison with composites with unmodified CNTs (3.81 ± 0.11 MPa). However, the addition of CNTs had no significant effect on SIBS with high molecular weight (M_n_~70,000 g mol^−1^) with ultimate tensile stress of pure polymer of 11.62 MPa and 14.45 MPa in case of its modification with 1 wt% of CNT-DDA. Enhanced biocompatibility of nanocomposites as compared to neat SIBS has been demonstrated in experiment with EA.hy 926 cells. However, the platelet aggregation observed at high CNT concentrations can cause thrombosis. Therefore, SIBS with higher molecular weight (M_n_~70,000 g mol^−1^) reinforced by 1–2 wt% of CNTs is the most promising material for the development of cardiovascular implants such as heart valve prostheses.

## 1. Introduction

Cardiovascular implants such as heart valve prostheses, vascular and stent grafts, patches, occludes, vascular closure devices, etc. are in wide demand in modern surgical practice due to the growth of the global cardiovascular disease burden [1]. Such devices improve the quality of human life by prolonging the functioning of the body systems beyond their expected duration. Synthetic biomaterials should exhibit high mechanical strength, chemical and enzymatic stability, and high hemocompatibility [2]. This necessity stimulates the development of new synthetic biomaterials with improved mechanical properties and biocompatibility. Preparation of nanocomposites using well-known synthetic polymers remains one of the relevant research topics. Therefore, the development of new nanocomposites based on high-strength, biologically inert carbon based nanosized materials–carbon nanotubes (CNTs) can be one of the possible solutions [3].

Interest in CNTs remains high due to their unique chemical nature: CNTs are graphene sheets rolled into a cylindrical shape [4]; belonging to one of the three allotropic modifications of carbon, they are made of *sp*^2^-hybridised carbon atoms [5]. Such unique structure and cylindrical shape provide high strength of nanoparticles, which is hundreds of times higher than the strength of steel [6]. The introduction of CNTs into the macromolecular matrix strongly affects the nanocomposites’ structure and mechanical properties, generally, increasing the tensile strength [7,8,9]. The materials’ strength is particularly important in biomedical applications such as vascular replacements, patches, chordal, and heart valve materials [10]. The length of CNTs is typically much larger than their diameter, so they can be considered as one-dimensional structures [11]. This feature provides nanoparticles with flexibility and helps to retain the elasticity of the polymer during nanocomposite preparation [12]. Moreover, the development of CNT-based polymeric materials can solve the issues of thrombus formation and infectious complications associated with using synthetic implants in clinical practice [13].

One of the most common preparation methods for composites is a direct blending of CNTs and corresponding (co)polymer. Despite significant advances in polymer-CNT composite preparation, the microstructure control of such materials remains an issue due to the strong interaction between individual CNTs leading to the formation of agglomerations [14]. The quality of the dispersion and the nature of the CNT distribution in the polymer matrix play a central role in tuning the properties of the obtained nanocomposites. The negative factors can be minimized by ultrasonic dispersion and modification (functionalization) of CNTs [15]. Exposure to the ultrasound makes it possible to reduce the size of agglomerates; however, in a solution, CNTs reconfigure themselves to a new equilibrium state with low energy due to reagglomeration [14]. Prolonged sonication, however, may lead to a decrease of the length of nanotubes that may negatively affect the properties of the final nanocomposite [16]. Reagglomeration can be impeded by: (1) the presence of suitable functional groups on the CNT surface; (2) surfactants non-covalently bonded with the CNTs due to π-π interactions and Van der Waals forces [17]. Most surfactants are toxic to the body [18] and cannot be used in biomaterials for non-covalent interaction. The covalent bonding method provides the formation of stable and strong bonds between the modifying agent and the CNT surface. Covalent modification is often carried out in two steps: first, the oxidation of CNTs to obtain carboxyl groups, and then esterification or amidation reactions with hydroxyl or amino-containing compounds [15]. However, oxidation under harsh conditions may result in a reduction of the CNT length, disruption of the tubular structure and *sp*^2^-hybridization of carbon atoms, whereas mild oxidants do not allow the formation of a sufficient number of -COOH groups [19].

The properties of polymer nanocomposites can be improved by chemical functionalization which increases the uniformity and stability of CNT dispersions in solvents, and also affects the nature of polymer-CNT interphase interaction [20]. The selection of the appropriate modifying agent determines the reinforcing effect of the nanoparticles. Primary aliphatic amines such as octadecylamine (ODA) and dodecylamine (DDA) are widely used for these purposes [21,22,23]. The research results showed that the stability of CNT dispersions in organic non-polar or low-polar solvents increases upon modification with aliphatic amines, while nanocomposites based on functionalized CNTs are stronger and more rigid than pure polymer [21,22,23]. Choosing a specific polymer-CNT ratio, conditions of modification and dispersion in the selected solvent depends on the chemical structure of the investigated solvents and polymers, however, the available literature data are insufficient.

The chemical nature of the macromolecular matrix certainly plays an equally important role in the development of nanocomposites for medical application. In order for the polymer in nanocomposites to meet the safety requirements, the polymer must be resistant to enzymatic, oxidative, and hydrolytic effects in the body, must not trigger the inflammatory response and thrombus formation, and must possess acceptable mechanical properties [24]. Poly(styrene-*block*-isobutylene-*block*-styrene) (SIBS) is compliant with the above requirements. SIBS is actively used in clinical practice as a coating for coronary stents due to the unique structure of the macromolecular chain [25]. We and other teams have demonstrated that SIBS is safe to use as a material for the fabrication of polymeric heart valves [26]. However, some studies have revealed the disadvantages of this material, such as surface cracking and plastic deformation, insufficient strength, which precludes its application in medical devices subject to pulse loads [26,27,28]. The incorporation of CNTs into SIBS can improve the mechanical properties of the polymer and can help to maintain its high biocompatibility, however, the irregular distribution of CNTs in the SIBS volume remains a problem [29].

In this work, we have examined two approaches for the preparation of SIBS nanocomposites, which consist of: (i) using carbon nanotubes modified with dodecylamine and (ii) unmodified (neat) CNT*s*. The distribution of nanotubes in triblock copolymers of different molecular weights and the influence of the content of nanotubes on the mechanical properties, thermal stability and bio- and hemocompatibility was evaluated. We found that the increase of the CNT content results in gradual improvement of the mechanical properties of SIBS with low molecular weight (Mn = 40,000 g mol^–1^) but the addition of CNTs had no significant effect on high molecular weight SIBS (M_n_~70,000 g mol^−1^). Moreover, the increase in concentration of CNT leads to the increase in the hydrophilicity and biocompatibility (cells viability), however, the platelet aggregation observed at high CNT concentrations can cause thrombosis. Therefore, SIBS with higher molecular weight (M_n_~70,000 g mol^−1^) reinforced by 1–2% of CNTs is the most promising material for the fabrication cardiovascular implants such as heart valve prostheses.

## 2. Materials and Methods

### 2.1. Synthesis and Characterization of SIBSs

Two samples of SIBS with different composition were obtained according to our previously developed method [29]. The molecular weight characteristics of the medium polyisobutylene block and final SIBS were determined by gel permeation chromatography using an Ultimate 3000 instrument (Thermo Fisher Scientific Dionex, Sunnyvale, CA, USA) equipped with a PLgel pre-column (7.5 × 50 mm, particle size 5 μm, Agilent Technologies, Santa Clara, CA, USA) and PLgel MIXED-C column (7.5 × 300 mm, particle size 5 μm). Detection was performed by refractometric and spectrophotometric (255 nm) detectors and tetrahydrofuran (LiChrosolv^®^, Merck, Darmstadt, Germany, >99.9%) was used as an eluent at 1 mL/min rate. Polymer’s M_n_ and M_w_/M_n_ values were calculated by using EasiCal (Agilent Technologies) polystyrene standards (580–400,000 g/mol and M_w_/M_n_ ≤ 1.05) and Chromeleon 7.0 software (Thermo Fisher Scientific Dionex). The ^1^H-NMR spectra of polymer solutions in CDCl_3_ (MagniSolv™, Sigma Aldrich, St. Louis, MO, USA, 99.8 atom% D, contains 0.1% (*v*/*v*) TMS) at concentration of ~15 g/L were measured at 25 °C on an AC-500 instrument (Bruker, Billerica, MA, USA) operating at 500 MHz.

### 2.2. Modification of CNTs

In the course of this study, single-walled CNTs with a diameter of 1.6 ± 0.4 nm, length more than 5 µm, and carbon content ~93% in CNTs (Sigma-Aldrich, St. Louis, MO, USA) were used. Sulfuric acid (98%, Panreac, Barcelona, Spain), nitric acid (70%, Sigma-Aldrich, St. Louis, MO, USA), dodecylamine (DDA, 98%, Sigma-Aldrich St. Louis, MO, USA) were used for the modification of CNTs.

Modification of CNTs was carried out according to the methodology developed by Ferreira et al. [22]. CNTs (0.4 g) were treated with sulfuric and nitric acids mixture (75 and 25%vol., respectively, 60 mL) in an ultrasonic cleaner for 6 h at room temperature. The resulting mixture was filtered under vacuum and washed with double distilled water until the pH of the water washes was neutral. Oxidized CNTs (CNT-COOH) were dried at 60 °C for 24 h. Then CNT-COOH (0.1 g) was distributed in 75 mL of DDA and the mixture was heated to 100 °C for 4 h with stirring. The modified nanoparticles (identified as CNT-DDA) were washed with ethanol to remove excess DDA and dried at 60 °C for a day. The generalized process of the CNT modification is shown in Figure 1.

### 2.3. Characterization of the CNT Structure

The chemical structure analysis of unmodified and modified CNTs was carried out by Fourier transform infrared (FTIR) spectroscopy using a Infralum FT-801 instrument (Lumex, Saint Petersburg, Russia) in a range of 400–4000 cm^−1^ with 4 cm^−1^ resolution in the KBr pellet and by Raman spectroscopy using Renishaw Invia Raman Microscope (Renishaw plc., Wotton-under-Edge, UK) equipped with an argon laser (λ = 514.5 nm) in a range of 400–4000 cm^−1^.

Transmission electron microscopy (TEM) was implemented for the nanoparticles crushed and subjected to ultrasonic treatment in acetone for 5 min by using a microscope JEM 2100 (JEOL, Tokyo, Japan) at 200 keV electron beam energy. CNTs were applied onto a preparative film by immersion and dried in air at room temperature to remove the traces of solvent.

### 2.4. Synthesis and Analysis of CNT Dispersions in Chloroform

CNTs, CNT-COOH, and CNT-DDA (0.1 g) were dispersed in chloroform (10 mL, >99.9%, Merck, Darmstadt, Germany) using an automatic UD–20 ultrasonic disintegrator (Techpan, Bialystok, Poland) with an output power of 180 W, a frequency of 22 kHz, and an oscillation amplitude of 8–16 µm for 1.5 h. A 40 μL drop of each dispersion was placed on a glass slide followed by air drying. The brightfield imaging of glass slides was performed using an AXIO Imager A1 microscope (Zeiss, Jena, Germany) at ×1000 magnification. The stability of the dispersions was visually determined after two weeks at room temperature. The particle sizes and size distributions of the CNT dispersions were observed using the Zetasizer dynamic light scattering (DLS) technique (Model: NanoZS90, Malvern Instruments Ltd., Malvern, UK) at scattering angle of 173° in a glass cell at temperature 20 °C. The number of measurements for each sample was at least 1000, until at least 10 convergent results were obtained. Correlation curves were calculated using the multiple correlation coefficient.

### 2.5. Preparation of SIBS-CNT Nanocomposites

CNTs and CNT-DDA were dispersed in 5 mL of chloroform using an automatic ultrasonic disintegrator UD–20 (Techpan) with an output power of 180 W, frequency of 22 kHz, and oscillation amplitude of 8–16 μm. CNTs were treated by ultrasound for a total of 1.5 h. After that A 5% *w*/*v* solution of SIBS in chloroform (5 mL) was added to dispersed CNTs followed by another round of sonication for 1.5 h. The resultant dispersion was cast and air-dried at room temperature for 24 h followed by air drying using an Emitech SC 7640 sputter coater (Quorum Technologies, Lewes, UK) at <2 × 10^−2^ mbar. The nanoparticles content varied within 1–8 wt%. The chemical composition of resultant samples was determined using FTIR spectroscopy.

### 2.6. Microscopy

The resulting nanocomposites were fractured in liquid nitrogen, and then the structure of the cleavage was analyzed by scanning electron microscopy (SEM). The studied samples were fixed on special tables and a conductive (Au-Pd) coating was formed on the surface by an ion sputtering method using an Emitech SC 7640 vacuum post (Quorum Technologies). Structural analysis of biomaterials was performed using a S-3400N scanning electron microscope (Hitachi, Tokyo, Japan) under high vacuum at an accelerating voltage of 5.0 kV in the secondary electron mode.

### 2.7. X-ray Microtomography (Micro-CT)

To evaluate the distribution of CNTs within SIBS matrix, we performed a micro-CT imaging of SIBS–CNT films using a custom-built CT setup with a micrometer resolution (voltage 80 kV; current 48 µA; film thickness 2.54 µm). The obtained grayscale CT scans were imported into the Avizo Fire 9 software (Thermo Fisher Scientific, Waltham, MA, USA), and a representative sample area of 500 × 500 × 25 µm^3^ was cropped for each sample, followed by a 3D reconstruction of X-ray density distribution.

### 2.8. Thermal Analysis

The thermogravimetry (TG) and differential thermogravimetry (DTG) were carried out in an STA 449 F3 Jupiter^®^ thermal analyzer (Netzsch Geratebau GmbH, Selb, Germany). Samples of about 3–4 mg mass were heated in a platinum crucible from 25 up to 600 °C in nitrogen atmosphere at a heating rate 5°/min.

### 2.9. Mechanical Tests

Uniaxial tensile testing was performed in accordance with ISO 37:2017 standard using a Z-series universal testing machine (Zwick/Roell, Ulm, Germany) equipped with a 50 N rated force sensor at 37 °C. Nanocomposites like control polymer samples were classified as isotropic materials taking into account the disordered distribution of reinforcing CNTs. Polymer films were cut using a dumbbell-shape knife (ISO 37:2017) with a width of 2 mm and length of 10 mm (n = 8 per group). Samples were subjected to one loading cycle with a constant speed of 50 mm/min until rupture. Ultimate Tensile Strength (MPa) was defined as a maximum load based on the initial samples’ cross-sectional area. The deformation properties were evaluated by the relative elongation, corrected by the nature of the destruction of the samples (%) and Young’s modulus (MPa), which remained within the range of small deformations (0.4–0.6 MPa).

### 2.10. Water Contact Angle Measurement

Water contact angle was determined by Drop Shape Analyzer DSA25 (Kruss GmbH, Hamburg, Germany) and sessile drop technique. A 15 μL drop of bidistilled water was spread on the flat surface of the samples and stabilized for 5 s for each measurement. All measurements were carried out at room temperature. The water contact angle was calculated from the results of five replicate measurements for each type of sample.

### 2.11. Cytotoxicity Assessment

The Ea.hy 926 cells (University of Northern California, Petaluma, CA, USA) were selected as an experimental endothelial cell line. The EA.hy 926 cell is a hybridoma line derived from human endothelium and A549/8 cells that demonstrate the main phenotypic and functional features of the endothelium in case of high proliferative activity. The cells were cultured in DMEM/F12 (11320033, Thermo Fisher Scientific), supplemented with 10% fetal bovine serum (26140079, Thermo Fisher Scientific), penicillin-streptomycin-glutamine solution: penicillin at 50 Units/mL, streptomycin at 50 µg/mL of, L-glutamine at 2 mmol/L (10378016, Thermo Fisher Scientific), amphotericin B (15290018, Thermo Fisher Scientific) and HAT Media Supplement (H0262, Sigma Aldrich). The passage in the culture was carried out upon 70% confluence. The cells were removed from the surface with a trypsin-EDTA solution at 0.025% (15400054, Thermo Fisher Scientific). The experiments were performed under sterile conditions, the cells were cultured in a CO_2_ incubator with 5% CO_2_ and high humidity. The Ea.hy 926 cells at 15 thousand cells/well were seeded on the matrices and cultured in a nutrient medium for 3 days. The control group consisted of samples without matrices in a 24-well plate, the sample with the appropriate number of cells was seeded and cultured under similar conditions. After 3 days, cell viability was assessed by means of fluorescence microscopy (n = 4, for each type of material) and metabolic activity was assessed by colorimetric analysis (n = 5).

To assess the viability, the cells were stained with Hoechst 33342 (10 µg/mL, 14533, Sigma Aldrich) for 10 min, and ethidium bromide (30 µg/mL, 46067, Sigma Aldrich) for 1 min. The cells in five randomly selected fields were counted with the Axio Observer Z1 microscope (Carl Zeiss). The results were presented as the number of cells per 1 mm of surface. Cell viability was calculated as the ratio of the difference between the total number of all adhered cells and the total number of dead cells to the total number of all adhered cells.

Cell Cytotoxicity Assay Kit (Colorimetric) (112118, Abcam, Cambridge, UK) was used to assess metabolic activity. The reagent was introduced into the wells with samples in nutrient medium at 1:5 concentration, incubated for 3 h at 37 °C. Following that, 200 µL of reagent from the sample were transferred to a 96-well plate, the optical density was measured at 570 nm and 605 nm wavelengths using the Multiskan Sky Microplate spectrophotometer (Thermo Fisher Scientific). The metabolic activity of cells was calculated via the difference in the ratio of the optical density of the sample at the wavelength of 570 nm to the optical density at the wavelength of 605 nm (OD_570_/OD_605_).

### 2.12. Platelet Adhesion

The hemocompatibility of polymer films assessed by the degree of platelet adhesion and the platelet transformation was characterized using scanning electron microscopy at 10 kV.

All donors provided a written informed consent before participation in the study. The study was conducted in accordance with the Declaration of Helsinki, and the study protocol received approval by the Ethics Committee of the Research Institute for Complex Issues of Cardiovascular Diseases (ID 0422021 issued on 12 October 2021). 

Samples of polymer matrices (n = 5) sized 0.5 cm^2^ each were incubated for 2 h at 37 °C in 300 μL of platelet-rich plasma (PRP) obtained from fresh citrated donor blood by centrifugation for 10 min at 1200 r/min. Next, the samples were washed with phosphate-buffered saline (PBS, pH-7.4) in order to remove non-adherent plasma components. Then the samples were fixed in a 2% glutaraldehyde solution for 24 h, washed with PBS and dehydrated in alcohols with concentration ranging from 30% to 100% for 15 min each, followed by drying at room temperature. Later, the samples were mounted on special tables using carbon tape to form a conductive (Au/Pd) coating using EM ACE200 vacuum coater (Leica Microsystemes GmbH, Wetzlar, Germany). Eight most exemplary samples were randomly selected for the analysis. The platelet adhesion was assessed by means of the platelet count (platelet per 1 mm^2^), the predominant type of platelets on the surface and by the platelet deformation index.

Platelet types depending on activation:Disc-shaped platelet, no deformation;Large platelet with pseudopodia-like protrusions;Large irregularly shaped platelet with pseudopods, platelets accumulate in conglomerates;Spreading platelet, the cytoplasm spreads and fills the spaces between pseudopods;Platelet dense granules, pseudopodia cannot be found due to the proliferation of the cytoplasm.

### 2.13. Statistical Analysis

Statistical analysis was performed using GraphPad Prism 7.0 (GraphPad Software, version 7.00, San Diego, CA, USA). The normality of distribution was assessed using the Kolmogorov-Smirnov test. The statistical significance of the differences between the groups was determined on the basis of variance analysis using Fisher’s parametric test, as well as post-hoc comparison. Differences between groups were evaluated using the nonparametric Kruskal-Wallis test with Dunn’s correction for multiple comparisons. In the case of normal distribution, the results were presented as the mean and standard deviation of the mean; in the case of non-normal distribution as the median, 25th and 75th percentiles (Me [25%; 75%]). *p* < 0.05 was the critical value for statistical significance.

## 3. Results

### 3.1. Synthesis of SIBS Copolymers

In this study, two samples of SIBS were prepared by the sequential cationic polymerization of isobutylene and styrene with dicumyl chloride/TiCl_4_/2,6-lutidine initiating system following the protocol mentioned in our previous study [30]. Two triblock copolymer samples with similar content of styrene in the copolymer but different number-average molecular weight and, therefore, different length of both of blocks were prepared (Table 1, Appendix A). Low molecular weight SIBS (SIBS-1) is characterized by a monomodal molecular weight distribution, while the high molecular weight copolymer (SIBS-2) showed small shoulder in high molecular weight region due to the intermolecular alkylation [30]. The complete shift of SEC traces into high molecular weight region upon addition of styrene to living polyisobutylene chains confirms the successful formation of triblock copolymers in both cases (Appendix A). The content of styrene in copolymer was calculated from the corresponding ^1^H-NMR spectrum (Appendix A) through the comparison of relative intensities of aromatic and aliphatic protons.

### 3.2. Characterization of the CNT Structure

The FTIR spectra of all studied samples (CNTs, CNT-COOH, and CNT-DDA) contain peaks at 1538 cm^−1^ due to the vibration of carbon atoms in the plane of the CNT graphene structure (Figure 1a) [31]. Stretching vibrations of O-H groups are presented as a broad signal in the spectral region about 3440 cm^−1^ (Figure 1a) [31]. Oxidation of CNTs leads to the appearance of an absorption peak at 1740 cm^−1^, which is specific to the stretching vibrations of the aliphatic carbonyl C=O in aldehyde or carboxylic acid groups [32]. Another peak noted in the CNT-COOH spectrum at 1380 cm^−1^ corresponds to stretching vibrations of the C-OH bond of the carboxyl group [33]. A significant increase in the intensity of the peak specific to stretching vibrations of C-H_x_ bonds at 2849–2920 cm^−1^ was found for nanotubes modified with dodecylamine CNT-DDA (Figure 1a) [34,35]. The intensity of the peak increasing at 1630–1640 cm^−1^ in the CNT-DDA spectrum is due to the stretching of the amide carbonyl N C=O (1635 cm^−1^, amide I), and also the vibration of unconjugated carboxyl-carbonyl groups [36] which are overlapping with the signal at 1630 cm^−1^ of the C=C aromatic ring bond [37]. The peak at 1630 cm^−1^ was observed for all samples. Strengthening the intensity and peak width in the CNT-DDA spectrum at 1538 cm^−1^ illustrates C-N stretching with C-N-H bending vibrations (amide II) [38]. The appearance of amide signals in the CNT-DDA spectrum not only confirms the grafting of DDA to the nanotube surface but also the hypothesis regarding the reaction mechanism (Figure 1). An intensive peak at 1460 cm^−1^ is associated with asymmetric bending vibrations of methyl groups -CH_3_ [39].

The Raman spectra of the studied samples (Figure 1b) contains: (i) radial breathing modes (RBM, ~200 cm^−1^); (ii) defects of the graphite layer (D-band, ~1350 cm^−1^); (iii) vibrations of carbon atoms in the graphite plane (G-band, ~1595 cm^−1^) split into low-frequency (G^−^) and high-frequency (G^+^) components; (iv) series of overtones of main vibrations [40]. The appearance of the G^−^ and G^+^ bands is due to the different energies of the longitudinal and transverse vibrations of the atoms in the presence of only *sp*^2^-hybridized carbon atoms of the graphite layer rolled into a tube. Thus, G^−^ and G^+^ are transverse and longitudinal vibrations, respectively. The analysis results show that the most ordered structure can be observed in the initial samples. Subsequent modification leads to structural defects in the graphite layers. Oxidation unambiguously reduces the order in the system, probably due to the destruction of the bonds within the graphene lattice. The following modification with DDA improves the ordering in the sample, but the sample becomes more defective than the initial CNTs. TEM images of nanoparticles showed changes in the surface structure of nanotubes after modification (Figure 1c). A significant amount of “amorphous carbon” after functionalization in the CNT-DDA samples was observed as well as an increase in the diameter and roughness of the outer surface of the tube, which indicates the quantitative functionalization of CNTs.

### 3.3. Analysis of CNTs Dispersions in Chloroform

Dispersions of each type of CNTs in chloroform were uniform and stable throughout visual analysis immediately after ultrasound exposure and 14 days after exposure. Nevertheless, when studying droplets of dispersions of unmodified CNTs on glass, clusters of macroscopic nanotubes were observed, while samples modified with DDA showed an improvement in the quality of particle distribution in the volume of the solvent according to optical analysis (Figure 2a).

CNT dispersions are characterized by the multimodal distribution with a set of 4–5 peaks and peak maxima in the range from 0.80 to 1000 nm (Figure 2b, Appendix A). The smallest peak corresponds to the average CNT diameter (diameter about 1.2 nm), the rest correspond to the aggregates, consisting of approximately 8 and 11 tubes. No peaks were detected in the region of micron sizes, which indicates a fairly high degree of dispersion of the CNTs, CNT-COOH, and CNT-DDA in chloroform.

It was found that the average diameter of the nanotube increases after the modification, and the oxidative modification has a greater effect. An increase in diameter indicates the formation of functional groups on the CNT surface. TEM analysis revealed a similar trend (Figure 1).

### 3.4. Analysis of the Structure of the Obtained Nanocomposites

Neat SIBS analyzed by SEM confirms the characteristic pattern of block-copolymers due to the microphase separation [41,42] (Figure 3a). The CNTs sheaves of different sizes were found in the fracture surface of nanocomposite films (Figure 3a). The size of such structures varied from 30 to 100 nm for different samples, thinner fibers were found in CNT-DDA-based nanocomposites as compared to SIBS-CNT. The increase in the weight content of nanoparticles in the polymer matrix results in the increase of the number of carbon nanoparticles in the volume of the polymer as well as of the size of aggregates. Evidently, the number of aggregates is higher for unmodified CNTs. It was shown that the nanocomposites change their X-ray density unequally with an increase in the nanofiller content. No significant increase in the nanofiller content was found in the SIBS-CNT-DDA samples, while in the SIBS-CNT samples the dependence is directly proportional. This result was confirmed qualitatively and quantitatively: (i) Figure 3b shows a slight change in the color of the SIBS-CNT-DDA samples with an increase in wt% of CNTs and insignificant increase in the (ii) the weighted average X-ray density in the sequence from 63.8 (0 wt%) to 65.7–69.9 units (1–8 wt%).

A detailed analysis of SEM images in the nanocomposites thickness reveals fine structures that differ in properties from the polymer matrix, the dimensions of which correspond to the dimensions of individual nanotubes (Figure 4a). It is also important to note that no nanotubes were found on the surface of nanocomposites. It means that the fiber-like elements are “packed” in a macromolecular matrix (Figure 4b).

The SIBS-CNT-DDA samples showed lower dependence of the X-ray density on the filler concentration compared to the original CNTs which indicates a more uniform distribution of the filler in the SIBS volume. On the contrary, SIBS-CNT samples change their structure more significantly: there are many areas of high density. In addition, the significant shift in the weighted average density from 63.8(0 wt%) to 70.4(1 wt%) and 79.3(8 wt%) was observed (Figure 5). In the present study, single agglomerates of high density were observed in the volume of the samples, for example, for concentrations of CNTs of 4 wt% and 8 wt%. However, they did not have a critical effect on the overall distribution of X-ray contrast such as the presence of defects in the histogram and its splitting (Figure 5).

### 3.5. Thermal Stability Evaluation

The TG curves (Appendix A) of composite films and neat triblock copolymer are characterized by one-step mass change that corresponds to the breaking of the polymer backbone and leads to the complete decomposition of SIBS. The temperature of the maximum decomposition rate (T_max_), initial decomposition temperature (T_onset_), and final decomposition temperature (T_end_) were estimated from TG and DTG curves and shown in Appendix A. The addition of CNTs and CNT-DDA into the SIBS polymer matrix increases the thermal stability of the polymer by significantly shifting T_onset_, T_max_, and T_end_ to higher temperatures. The minimum and maximum T_onset_ values were obtained for the pure SIBS polymer (365.9 °C) and for the CNT-DDA-8% composite (377.7 °C), respectively. The maximum T_onset_ value is several tenths of a degree lower for a nanocomposite containing 8% unmodified CNTs (377.5 °C). The general trend is the gradual increase in thermal stability with an increase in the nanoparticles content, as well as higher stability of SIBS-CNT-DDA samples in comparison to SIBS-CNT with the same amount of nanofiller.

### 3.6. Mechanical Tests

The mechanical properties of SIBS differed significantly depending on the molecular weight (Figure 6a–d). Moreover, the rigidity and strength of SIBS depended on CNTs and CNT-DDA concentration (Figure 6a–d).

Initially, SIBS-2 with high molecular weight was six times stronger than SIBS-1, showed higher percentage of ultimate strain (28%, *p* < 0.01) and did not differ in Young’s modulus (*p* = 0.99) (Figure 6e–g). This difference in tensile strength is consistent with the higher length of polystyrene side blocks in SIBS-2 (M_n_~15,000 g mol^−1^) as compared to SIBS-1 (M_n_~8700 g mol^−1^) [30,43]. Since SIBS belongs to the thermoplastic elastomer family, the increase in the length of “hard” polystyrene block results in the enhancement of interaction between polystyrene blocks of different polymer chains in polystyrene domains that, in turn, leads to the increase of tensile strength.

Increasing CNTs concentration in the nanocomposites resulted in increased strength of SIBS-1 with low molecular weight (nanofiller content of 4 wt%, *p* < 0.05), while the strength of SIBS-2 with high molecular weight did not differ upon changing the percentage of nanofiller. Following the incorporation of 8 wt% CNTs, the strength of SIBS-1 polymer increased 5-fold. For both types of nanocomposites, incorporating over 2 wt% of nanofiller significantly reduced the ultimate strain (*p* < 0.05) and elastic (Young’s) modulus (*p* < 0.05) of the material. The strong influence of the content of CNT*s* on tensile strength in SIBS-1 could be explained by the interaction of nanotubes with polystyrene blocks of SIBS. In case of SIBS-2 with longer polystyrene side blocks the weak Van der Waals/electrostatic interactions between CNT*s* and polystyrene blocks in triblock copolymer, probably, do not exceed the Van der Waals interactions between different polymer chains. Therefore, the introduction of CNTs had a little effect on tensile strength in this case and even led to its decrease.

Comparing the dynamics of changes in the strength between polymers with modified and unmodified CNTs revealed the presence of a 5% barrier. Keeping CNT-DDA concentration within this barrier makes it more effective than unmodified CNTs. Modification with DDA functionalized nanotubes did not positively affect the mechanical properties of SIBS-1 with low molecular weight (as compared to SIBS-1-CNT), with the exception of 4 wt% SIBS-1-CNT-DDA, which demonstrated a 135% increase in ultimate tensile stress as compared to SIBS-1 (*p* < 0.05) and a 45% increase as compared to 4 wt% SIBS-1-CNT (*p* < 0.05). At the same time, incorporating CNT-DDA into 2 wt% SIBS-2 with high molecular weight increased ultimate strain (*p* < 0.05), and incorporating CNT-DDA into 6 wt% SIBS-2 reduced the rigidity (*p* < 0.05) in comparison to unmodified SIBS-2-CNT with similar CNTs concentration. 2 wt% SIBS-2-CNT-DDA exhibited ultimate tensile stress higher by 20% as compared to the unmodified SIBS-2 (*p* < 0.05) and ultimate tensile stress higher by 38% as compared to 2 wt% SIBS-2-CNT (*p* < 0.05).

SIBS-2 with high molecular weight and without nanofiller provides ultimate tensile stress of 11.62 MPa, ultimate strain of 726%, and Young’s modulus of 0.77, demonstrating the prospects of its application in medical devices that require high elasticity, strain density, and a margin of safety. At the same time, the modification of 1wt% SIBS-2-CNT-DDA leads to an increase of the ultimate tensile stress to 14.45 MPa (*p* < 0.01) while maintaining its elastic modulus (*p* = 0.99). Subsequent increasing of CNTs concentration improves the rigidity of the material but does not lead to an increase in its tensile strength.

### 3.7. Evaluation of Cytotoxicity and Wettability

The hydrophilicity of the samples estimated by the water contact angle measurement proportionally increases with the content of carbon filler for both types of SIBS-CNT and SIBS-CNT-DDA nanocomposites (Figure 7). Neat SIBS has the water contact angle of 94.06 ± 1.69° and can be classified as a hydrophobic material. The SIBS-based nanocomposites are more hydrophilic in comparison with pure SIBS and have water contact angles from 88.50 ± 4.43° to 50.46 ± 2.56° for 1 wt% SIBS-CNT and 8 wt% SIBS-CNT-DDA, respectively.

An increase in the nanotubes content in the SIBS-CNT and SIBS-CNT-DDA composites to 6 and 8 wt% leads to an increase (*p* < 0.05) in the number of EA.hy926 cells adhered to the surface and their viability (*p* < 0.05) (Figure 8a,b). At the same time, the values reflecting the number and the viability of cells in the indicated samples and culture plastic were similar (*p* = 0.62). Initial polymer and its nanocomposites with a nanofiller concentration of 1% moderately disrupt cell activity simultaneously reducing cell adhesion and viability (Figure 8a,b). On the contrary, 1 wt% SIBS-CNT-DDA sample favorably differs from a similar composite based on unmodified CNTs because the number of cells on its surface is significantly higher (*p* < 0.05) and the cell viability tends to be higher as well. 6 wt% SIBS-CNT-DDA nanocomposites also have an advantage (*p* < 0.05) over 6 wt% SIBS-CNT samples in terms of the number of EA.hy926 cells adhered to the surface. At the same time, both types of composites with a filler content of 6 and 8 wt% do not reduce cell viability since the culture plastic exhibited similar values (Figure 8b). The adhesion and viability of EA.hy926 cells on the surface of pure CNTs and CNT-DDA (100 wt%) do not differ (*p* = 0.86) between each other and between samples of the corresponding nanocomposites containing 0, 1, 2, and 4 wt% nanoparticles in the composition.

The metabolic activity of cells on the samples containing 6 and 8 wt% CNTs and CNT-DDA is higher (*p* < 0.05) compared to 0 and 1 wt% in the corresponding group (Figure 8b). There were no differences (*p* = 0.83) in the metabolic activity of EA.hy926 cells on culture plastic and on the 6 wt% SIBS-CNT, 8 wt% SIBS-CNT, 4 wt% SIBS-CNT-DDA, and 6 wt% SIBS-CNT-DDA samples. These data are coherent with the obtained data on viability and the number of cells on the samples’ surface.

### 3.8. Platelet Adhesion

All studied samples exhibited adequate platelet-surface adhesion, but differed in platelet spreading. Unmodified SIBS was characterized by a total uniform platelet spreading on the surface, while platelets on nanocomposite surfaces were aggregated and spread fragmentally in materials containing higher content of nanofillers (Figure 9).

Types III and IV platelets were the most predominant on the surface of SIBS, while type V platelets and type II platelets were found less frequently. Incorporating 1 wt% CNTs into SIBS did not affect the platelet adhesion. Increasing CNTs concentration to 2 wt% led to a slight decrease in the platelet count (*p* < 0.05) and in the platelet deformation index (*p* < 0.05). SIBS films with 6 wt% CNTs (SIBS-CNT-6%), films with 1 wt% DDA functionalized CNTs (SIBS-CNT-DDA-1%), and unmodified SIBS were characterized by the predominance of types III and IV platelets, and fewer V type platelets. The predominance of type III platelets was typical for films with 1 and 2 wt% CNTs, type IV–for films with 4 and 8 wt% CNTs and most films containing DDA functionalized CNTs (except SIBS-CNT-DDA-1%). It is worth noting that incorporating 1 or 8 wt% DDA functionalized CNT nanoparticles into SIBS polymer matrix led to a decrease in the platelet adhesion as compared to unmodified CNTs with similar concentration (*p* < 0.01 and *p* < 0.05, respectively).

Both 2 wt% SIBS-CNT and 1 wt% SIBS-CNT-DDA nanocomposites differed favorably from the rest of the studied materials, including unmodified SIBS (*p* < 0.05). An increase in the concentration of nanofillers in SIBS polymer matrix resulted in higher thrombogenicity of the material.

## 4. Discussion

The fabrication of polymer nanocomposites with CNTs for biomedical applications is of high interest due to the unique properties of CNTs such as high strength, unique geometry, and flexibility [40,44,45]. However, the tendency of CNTs to aggregate in organic solvents and polymer solutions/melts necessitates further studies to search for new approaches to the production of nanocomposite biomaterials. The common approach for suppressing the agglomeration of CNTs is the use of surfactants, which are attached to surface of nanotubes thus preventing the agglomeration or even leading to deagglomeration [46,47,48,49]. Nevertheless, most of these surfactants are toxic to the body and can be released during the positioning of the implant, thus limiting their application in the preparation of biocompatible nanocomposites. Therefore, in this work we covalently attached DDA (a surfactant) to the CNT surface through the amino group according to the Firerra method [22,50]. To do that, CNTs were subjected to oxidation, followed by esterification or amidation reactions with hydroxyl or amino-containing organic molecules [15].

The success of the covalent grafting of DDA through the amino group was confirmed by FTIR spectroscopy (Figure 1a), especially by the presence of the amide bands, which is in agreement with data from [50]. TEM was utilized to confirm the successful grafting of ODA to the CNT surface [23,51]. The TEM images obtained in the present study indicate a change in the CNT structure and the distribution of DDA at the edges of the tubes and on the outer side of the CNT wall (Figure 1c). The chloroform used in this study dissolves SIBS very well and was easily removed by evaporation, which is useful for films prepared by polymer solution casting and for medical application as well [52]. It should be noted that optical microscopy indicates the significant improvement in the distribution of modified particles (CNT-DDA) in comparison with unmodified ones (CNT) (Figure 2a). These observations were also confirmed by DLS measurements showing the formation of smaller aggregates in case of modified carbon nanotubes (see Figure 2b and discussion therein).

Composite films were obtained by incorporating SIBS to CNT or CNT-DDA dispersions in chloroform and subsequent polymer solution casting. The stability of CNT dispersions in polymer solutions does not deteriorate [53], making it possible to fabricate nanocomposite materials that are visually homogeneous. Usually, “wrapping” occurs at the interface between carbon nanotube and polymer matrix, although the physicochemical states of these interfacial interactions remain poorly understood [45]. Supposedly, the electrostatic and van der Waals forces play a crucial role in them, while the hydrophobic interaction between the aliphatic chains of CNT-DDA and SIBS provides a smaller but significant contribution [45].

SEM analysis of the cross-section of the obtained films provides information on the CNT distribution in the polymer thickness [54]. Sample preparation upon fracture in liquid nitrogen allows preserving the molecular structure of the polymer nanocomposite [55]. The characteristic block copolymer pattern was observed both in pure SIBS and in the CNT-free regions of nanocomposites due to microphase separation (Figure 3a) [41,42] that indicates the preserving specific packing of the polymeric domains even after the reinforcing component addition. Fibrous structures that are visualized in the SIBS-CNT cross-section and have higher electron density (lighter in the SEM images) can be attributed to agglomerates consisting of several dozen individual tubes. The presence of such agglomerates with relatively smaller sizes in CNT-DDA nanocomposites is probably due to incomplete surface modification (Figure 3a). Nevertheless, the fiber sizes for both types of nanocomposites do not exceed 100 nm, which is less than the maximum sizes of agglomerates recorded by the DLS method in dispersion (Appendix A). As noted earlier, DLS allows specialists to determine only the average size of spherical particles which is not the diameter of the detected agglomerates. Individual nanotubes located in the thickness of the polymer matrix are almost undetectable in the secondary electron mode, but can be found in the images obtained by backscattered electrons mode. However, SEM does not provide the information about the 3D structure of the sample without violating its integrity, thus, it was used only to visualize a small surface layer or a thin sample. The instruments for non-destructive structure inspection within the material volume using high-resolution radiological methods are especially important for assessing the quality of CNTs distribution (Figure 3b) [56]. Micro-CT 3D images show a more uniform particle distribution in SIBS-CNT-DDA composites. An increase in the X-ray density for the series of SIBS-CNT samples was noted along with an increase of the concentration of nanoparticles and associated with the appearance of a larger number and size of agglomerates. The appearance of volume defects such as polymer-rich regions, CNTs, and agglomerated voids can provoke stress concentration in these regions and initiate crack growth [57]. The successful preparation of SIBS-CNTs nanocomposites is also confirmed by the shifting in the decomposition temperature towards higher temperatures with increase of nanotubes content that correlates well with the literature data (Appendix A) [54,58,59].

The dispersed in the polymer matrix CNTs can prevent the flow of decomposition products and thus delaying the onset of decomposition, which explains the increase in the thermal stability of composites with increasing nanofiller contents [60]. In addition, the shift of T_max_ towards higher temperatures may be due to the slower decomposition of polymer molecules that are in contact with much more thermally stable CNTs (Appendix A). An increase in the thermal stability of nanocomposites can also be driven by an increase in thermal conductivity leading to the heat dissipation inside the composite [61].

The tensile stretch of the SIBS polymer material is several times higher than that of biological material [25,62], however, the mechanical properties of copolymers investigated in this work significantly affected by the molecular weight. As we have demonstrated here, the length of side polystyrene blocks in SIBS is crucial to reach the required for application mechanical properties. SIBS-2 with long polystyrene blocks (M_n_~15,000 g mol^−1^) showed good mechanical properties and the addition of CNT*s* allowed mainly to tune the elongation and Young’s modulus due to strong enough Van der Waals interactions between polystyrene blocks. In contrast, the addition of CNT*s* into SIBS-1 possessing short polystyrene side blocks (M_n_~8500 g mol^−1^) results in significant improvement of tensile strength due to the stronger interactions of nanotubes with polystyrene blocks in comparison with weak Van der Waals interactions between short polystyrene blocks (Figure 6).

As it is evident from Figure 6, the effect of the content of CNT*s* on the ultimate strain and Young’s modulus is similar for both SIBS samples: the increase of the content of CNT*s* leads to gradual decrease of ultimate strain and gradual increase of Young’s modulus, respectively. This observation is consistent with the uniform distribution of CNT*s* in both polystyrene domains and polyisobutylene matrix as it is evidenced from Figure 3. The interactions of CNT*s* with polyisobutylene segments of SIBS are responsible for the observed change of ultimate strain and Young’s modulus.

Excessive deformation with insufficient material rigidity may cause significant deformation of the prosthetic valve or produce a potentially dangerous effect in the formation of graft aneurysms. A decrease in deformation upon fracture of nanocomposites (Figure 6f) can be explained by the ability of CNTs to prevent crack propagation and absorption of fracture energy due to interfacial deformation between nanoparticles and a polymer matrix [63]. It should be noted that polymer nanocomposites are usually resistant to cyclic loads due to their unique mechanical properties [64] which is a key requirement for the safe functioning of a heart valve prosthesis in the body for at least 250 million cycles. In similar studies [65] polyethylene-based composite with CNTs was found to be more resistant to cyclic loads than a pure polymer. The addition of nanotubes made it possible to improve of the mechanical properties of the SIBS polymer [29]: the films become more rigid, Young’s modulus and maximum tensile strength increased (Figure 6). Similar effect was observed for CNT-polydimethylsiloxane [66], CNT-polyethylene terephthalate [63] and CNT-thermoplastic polyurethane [8] nanocomposites, respectively.

CNT-DDA has a greater effect on the composite Young’s modulus of SIBS nanocomposites than CNTs which could be explained by the improved interfacial interaction after functionalization. Indeed, De Menezes et al. found an increase in Young’s modulus and tensile strength when ODA modified CNTs were introduced into polyethylene compared to nanocomposites based on pure CNTs, which was explained by the similarity between long chains of the ODA structure and polyethylene [23]. ODA like DDA has a long aliphatic chain with an amino group at the end which results in the similarity of the mechanism of action of DDA on mechanical properties. The initial increase of tensile strength with the addition of small amounts of CNTs followed by its decrease at higher concentration of CNTs observed for high molecular weight SIBS (SIBS-2, Figure 6e) is consistent with the literature data [67]. Such effect was explained by agglomeration of CNTs that can lessen the reinforcement effect of CNTs by ineffectively transferring the external load [67]. The higher tensile strength for composites containing CNT-DDA as compared to unmodified CNTs could be explained by better distribution of nanotubes in a polymer matrix [23]. The higher tensile strength of SIBS-CNT samples with 6 and 8 wt% nanoparticle contents as compared to SIBS-CNT-DDA is probably due to the formed bonds between bundles of carbon nanofibers.

The main disadvantage of CNTs in terms of biomedical applications is their possible toxicity to the human body cells. It is likely that CNTs penetrate cell membranes due to hydrophobic contact [68] since there are no immobilized hydrocarbon chains on the nanomaterial surfaces. Therefore, it is possible to reduce the negative effect of nanotubes on the body by functionalizing them [69]. For example, the cytotoxicity analysis of biocompatible polyurethane incorporated with 1% CNTs showed no negative effect on human umbilical vein endothelial cells (HUVEC) after 7 days of cultivation [62]. It was previously reported that SIBS-CNT films support the growth of L-929 fibroblasts [70] and do not adversely affect the HUVEC cells [29]. Our results showed an increase in the viability and metabolic activity of EA.hy926 cells on the nanocomposites (especially with 6 and 8%wt of CNTs) compared to neat SIBS and neat nanotubes that is primarily associated with the packing of fibers inside the polymer matrix (Figure 4) and growing hydrophilicity of materials (Figure 7). In the present study, the water contact angle of SIBS films decreased with the inclusion of CNTs into the polymer matrix (at 4, 6, and 8 wt%). The absence of significant differences between the SIBS-CNT-DDA and SIBS-CNT composites suggests that the chemical structure of the nanotube surface had fewer effects on cytotoxicity.

Nevertheless, we observed a difference in the water contact angle value between nanocomposites with different types of CNTs in their composition. The obvious reason for this difference is a change of the chemical structure of nanotubes after modification. In our case, another factor that has a greater effect on the hydrophilicity of the samples is the surface topography [71]. The change in the surface relief at the micro level leads to an increase in the hydrophobicity of hydrophobic materials and the hydrophilicity of hydrophilic materials [72]. Neat SIBS has a contact angle of 94.06°, intermediate between those of the polystyrene (86.0°) and polyisobutylene (103.9°) from which it is constructed [73,74]. The inclusion of nanoparticles into the polymer structure can result in not only the appearance of surface roughness but also in changes in the blocks arrangement (order), which together leads to an increase in hydrophilicity. The wide scatter of water contact angle values is probably due to the inhomogeneity of the surface structure. However, we observe a trend towards a gradual (linear) decrease in the water contact angle with the transition to composites containing more CNT (Figure 7).

Thrombosis is one of the most severe disadvantages of biological heart valves. Thrombosis can be caused by platelet disorders that lead to defects in primary hemostasis [75]. CNTs can induce platelet activation by penetrating cell membranes [76]. However, in some cases, incorporating CNTs into the polymer matrix does not increase the degree of platelet adhesion and platelet activation, leading to a decrease in these parameters instead [77,78]. For example, incorporating multi-walled CNTs has improved hemocompatibility of poly(lactic-co-glycolic-acid) and polyurethane [77,78]. The authors achieved these results by changing the chemical composition and surface relief of modified materials [77]. With certain concentrations, free CNTs distributed in nanocomposite have minimal effect on the environment. Thus, the findings regarding a slight decrease in thrombogenicity in films containing 1 and 2 wt% CNTs (Figure 9) were consistent with the results of other studies [77,78]. As we noted earlier, this observation may be due to an increase in the hydrophilicity of the material with a moderate increase in roughness. However, the increase of the content of nanotubes in the composition leads to an increase in the degree of platelet activation and the number of adherent platelets due to the incorporation of non-coated nanotubes to the surface of nanocomposites as well as the increase in surface roughness.

## 5. Conclusions

In this work, we assessed the influence of the method of CNT preparation and molecular weight of triblock copolymers of isobutylene with styrene on the mechanical properties, thermal stability and bio- and hemocompatibility of the obtained nanocomposites. We have demonstrated that the modification of carbon nanotubes by oxidation and subsequent amidation with DDA provides better dispersion of CNTs in SIBS as compared with neat (unmodified) nanotubes. The addition of nanotubes to SIBS matrix (up to 8 wt%) results in significant improvement of the mechanical properties of SIBS with low molecular weight (M_n_ = 40,000 g mol^–1^) leading to visible increase in tensile strength and Young’s modulus, and decrease of elongation at break. In contrast, the effect of CNTs on tensile strength of SIBS with high molecular weight (M_n_~70,000 g mol^−1^) is insignificant, while the increase of CNTs content in polymer leads to gradual increase in Young’s modulus and decrease of elongation at break, respectively.

The increase in CNTs content leads to the increase in hydrophilicity and biocompatibility (cells viability) of obtained SIBS nanocomposite. However, the platelet aggregation observed at high CNT concentrations can lead to the thrombosis formation. Therefore, SIBS with high molecular weight (M_n_~70,000 g mol^−1^) reinforced by 1–2 wt% CNTs is the most promising material for the fabrication of heart valves. Despite of this positive outcome, future research on the durability of the SIBS nanocomposites is needed before the translation of the research findings into clinical practice.

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
