# Peer review of "Biomaterials Based on Carbon Nanotube Nanocomposites of Poly(styrene-b-isobutylene-b-styrene): The Effect of Nanotube Content on the Mechanical Properties, Biocompatibility and Hemocompatibility"

_nanomaterials, 2022, doi:10.3390/nano12050733_

Round 1
Reviewer 1 Report
In this manuscript, Rezvova and coworkers present work about nanocomposites based on poly(styrene-block-isobutylene-block-styrene) (SIBS) with singlewalled carbon nanotubes (CNTs) which were prepared and characterized in terms of tensile strength as well as bio- and hemocompatibility. This manuscript presents some new results that will be of interest to the Readers of Nanomaterials mdpi after major revision. There are several concerns the authors need to address. These are listed below:
- On page 3, line 115, please explain, which two approaches are proposed? What do each of them mean?
- Part of Scheme 1 is taken from reference number 20 that was not noted by reference. Also, can primary amines interact with hydroxyl groups under such conditions?
- On page 10, line 386-387, the sentence is incorrect.
- On page 13, line 425 TGA is an important method that provides information about the nature of the interaction between the polymer matrix and nanotubes. Unfortunately but authors only summarize obtained results but appropriate disscusion is absents.
- In section 3.6. also no discussion. Only facts are given, but unfortunately there are no explanations.
- In section 3.7. describes the results of measuring the contact angle of wetting. The sentence on line 482 has no meaning. I would also like to read why there is no directly proportional change in the contact angle of wetting in the number of carbon nanotubes. No explanation or discussion.
- We recommend adding some missing references:
- https://doi.org/10.1016/j.apsadv.2021.100104
- https://doi.org/10.3390/ma11030383
- There are also minor corrections that need to be made to the manuscript. These include the following:
- On page 2, line 66 change “morevoer” to “moreover”;
- On page 4, line 154, line 173 change “ml” to “mL”;
- On page 15, line 475 put “,” after “Further”;
- On page 21, line 659 put “,” after “case”.
Therefore, I believe the target journal is an appropriate forum for this article after major revision.
Reviewer 2 Report
Carbon nanotubes have great potential as biomaterials. In this paper, the mechanical properties, biocompatibility and hemocompatibility of CNT-SIBS composites are investigated. By adding the optimum amount of CNTs, the biocompatibility and hemocompatibility were improved. These are useful findings for the bio-application of CNT composite materials. Therefore, I recommend that the paper is acceptable for publication after the following concerns are fixed.
1. In this paper, the CNT particle size in dispersion is analyzed by DLS. However, since the scattering intensity of DLS signal is proportional to the sixth power of the particle size, the presence of small particles is usually hidden by the scattering of large particles. So, it is hard to believe for me that precise and quantitative analysis is possible for CNT particles with broad size distribution. The quantitative analysis shown in Figure S1 should be supported by the other methods.
2. It is stated that the increase in hydrophilicity due to the inclusion of CNTs is associated with changes in the topology of the film (p.21). This should be explained in more detail in the text.
3. The reason for the slight decrease in thrombogenicity in films containing 1 and 2 wt% CNTs should be explained (p. 21).
Round 2
Reviewer 1 Report
The authors have addressed all my concerns properly. It can be published as it.Author Response
Dear Reviewer,
Thank you for your very careful review of our paper.
Reviewer 2 Report
The authors answered my questions, but I still doubt the result of the DLS analysis. For the particles having broad size distribution, DLS analysis sometimes provide wrong results. If authors want to claim that the CNTs mainly disperse as isolated forms, I recommend that they try the other method such as analytical centrifuge (see for example, M. Nadler et al, Carbon, 46 (2008) 1384–1392; M. S. Arnold et al., ACS Nano, 2 (2008) 2291–2300.).
